# Mental Health Experts as Objects of Epistemic Injustice—The Case of Autism Spectrum Condition

**DOI:** 10.3390/diagnostics13050927

**Published:** 2023-03-01

**Authors:** Maciej Wodziński, Marcin Moskalewicz

**Affiliations:** 1Institute of Philosophy, Maria Curie-Skłodowska University, M. Curie-Skłodowska sq. 4, 20-031 Lublin, Poland; 2Doctoral School of Humanities, Maria Curie-Skłodowska University, Weteranów 18, 20-038 Lublin, Poland; 3Philosophy of Mental Health Unit, Department of Social Sciences and the Humanities, Poznan University of Medical Science, Rokietnicka 7, 60-806 Poznan, Poland; 4Phenomenological Psychopathology and Psychotherapy, Psychiatric Clinic, Heidelberg University, Voßstraße 4, 69115 Heidelberg, Germany

**Keywords:** expert knowledge, testimonial injustice, hermeneutical injustice, discrimination, autism, schizophrenia, mental health diagnosis, social epistemology

## Abstract

This theoretical paper addresses the issue of epistemic injustice with particular reference to autism. Injustice is epistemic when harm is performed without adequate reason and is caused by or related to access to knowledge production and processing, e.g., concerning racial or ethnic minorities or patients. The paper argues that both mental health service users and providers can be subject to epistemic injustice. Cognitive diagnostic errors often appear when complex decisions are made in a limited timeframe. In those situations, the socially dominant ways of thinking about mental disorders and half-automated and operationalized diagnostic paradigms imprint on experts’ decision-making processes. Recently, analyses have focused on how power operates in the service user–provider relationship. It was observed that cognitive injustice inflicts on patients through the lack of consideration of their first-person perspectives, denial of epistemic authority, and even epistemic subject status, among others. This paper shifts focus toward health professionals as rarely considered objects of epistemic injustice. Epistemic injustice affects mental health providers by harming their access to and use of knowledge in their professional activities, thus affecting the reliability of their diagnostic assessments.

## 1. Introduction: Epistemic Injustice in Mental Health as concerning Both Service Users and Providers

Can only ‘patients’ or ‘service users’ of mental healthcare systems be objects of epistemic injustice? Can health professionals (traditionally portrayed as the perpetrators of unfairness) also suffer from such injustice in their interactions with patients? If so, what might be the sources of epistemic injustice understood in such an unorthodox manner? By answering these questions, this paper presents the issue of epistemic injustice from a novel perspective. We hypothesize and argue that health professionals may also become objects of epistemic injustice and that recognizing this may significantly impact their practice.

Miranda Fricker’s influential work has brought much attention to the injustice and harm performed on people concerning knowledge production and distribution. Discussing epistemic injustice, researchers and public opinion focused on people typically associated with the ‘weaker side’ in different settings of power relations. In healthcare systems, these were the patients. It is typically pointed out that such harm is caused by the “stronger” side of the interaction. For example, doctors use stereotypes concerning various groups of patients, experts reject (without proper justification) the opinions of other experts about the patient’s health, diagnosticians use biased diagnostic tools, or nurses, social workers, and even patients’ families ignore the first-person viewpoint of their ill members.

This paper shows that the ‘stronger side’ of these power relations might also be harmed. Considering the phenomenon of ‘being an expert’ from an epistemological and ethical perspective, one may notice that it involves not only the possession of specific competencies, but also the obligation to perform one’s duties diligently. In the case of mental health, this is both extremely important and challenging. With this in mind, this review draws attention to the fact that those usually perceived as the ‘stronger side’ can also fall victim to epistemic injustice.

Epistemic injustice, a concept developed by philosopher Miranda Fricker [1,2], is a situation in which, without adequate reason, someone suffers harm caused by or related to access to the process of producing and distributing knowledge. As Grim and colleagues point out, the notion of epistemic injustice can refer not only to the social practices involved in the production and transmission of knowledge, but also to the very status of subjects and their position in the social hierarchy of knowledge-making practices:

“The concept of epistemic injustice refers to an injustice performed on individuals in their capacity as knowledge bearers, reasoners and questioners, in which their ability to take part in epistemic practices, such as giving knowledge to others (testifying) or making sense of their experiences (interpreting), is weakened”.[3] (p. 158)

Table 1 presents the core arguments made further in this article.

## 2. Epistemic Injustice toward Patients

When considering the issue of epistemic injustice, among other things, one can refer to phenomena, such as epistemic exploitation [4], epistemic oppression [5,6], systemic silencing [7], genocide denial [8], or even hermeneutical death [9,10]. Although the literature describes many possible types of epistemic injustice, this paper focuses on its two main modes, described by Fricker, namely, testimonial and hermeneutical injustice. The former refers to a situation in which, without a sufficiently justified reason, the testimony of one subject is omitted or belittled (or its ability to carry out the relevant cognitive processes is denied): “Testimonial injustice” occurs when prejudice causes a hearer to ascribe a deflated level of credibility to a speaker’s words or ‘testimony’ [11] (p. 151). The second, hermeneutical injustice, refers to a situation in which the subject itself is ‘epistemically wronged’ due to not having resources sufficient for the ability to interpret its own experience adequately.

In recent years, a number of researchers considered the occurrence of epistemic injustice in healthcare systems, including mental health [12,13,14,15,16,17,18,19]. They pointed out that becoming ill exposes patients to the experience of such injustice and, consequently, to a lower chance of receiving appropriate help [20,21,22]. Lakeman emphasized that the causes of testimonial injustice tend to have individual origins and essentially lie at the heart of the mental health care system:

“Actual and potential testimonial injustice is endemic within mental health service delivery. For example, central to mental health legislation is the idea that some people lack the capacity to make decisions and it follows that what they might say, how they construe problems, their choices and preferences lack coherence, logic, or credibility. It is not surprising then that the testimony of all or most people who use mental health services might be considered suspect”.[11] (p. 151)

The second type of epistemic injustice, hermeneutical injustice, is more socially grounded. It is a social situation in which an individual has difficulty making adequate sense of his or her experience and interpreting it due to the so-called “collective hermeneutical block” or bias. Explaining the ‘social basis’ of hermeneutical injustice, Grim and colleagues note that:

‘Hermeneutical injustice occurs when there is a breach in shared conceptual, interpretative resources that puts individuals at a disadvantage when trying to make sense of their experiences’.[3] (p. 158)

An example given by Fricker is that women experienced sexual harassment at a time when the term did not yet exist. This is not, of course, just a lack of a label for certain behaviors but rather a set of social conditions (such as consent to certain behaviors, women thinking of themselves as inferior to men, cultural objectification of the body) that prevented women from thinking of themselves as victims of harassment.

A structurally similar situation can be observed for people reaching out to mental healthcare. The dominant narratives (social, medial, academic, and scientific) very strongly shape how the participants in this system see themselves (both in terms of how they perceive their characteristics and in terms of their position in the web of relationships between the different parties involved in healthcare) [23,24,25,26]. Epistemic inequalities result in ethnic minorities being forced to use crisis care and being less likely to receive primary mental health care [27]. The differences between patients’ first-person perspectives on illness and service providers’ conceptualization of illness as a disease are growing [12]. Patients’ testimonies of perceived illness symptoms are downgraded [28], and somatic illnesses are often overshadowed by previously detected mental conditions leading to a decrease in the quality of medical care [29]. Also, epistemic injustice impacts the policy-making level, resulting in the exclusion of specific patients or health professional groups from the legislative processes [30,31].

Exploring the ideas such as epistemic solidarity among patients or increasing the role of biocommunities in knowledge production and circulation might reduce epistemic injustice in mental health [32,33]. Equally important is the identification of causes and additional factors influencing the occurrence of epistemic injustice, such as the hubris phenomenon [34,35] or low levels of intellectual humility among professionals [36,37], which increase the risk of systemic cognitive errors.

## 3. Epistemic Injustice toward People on the Autism Spectrum

People on the autism spectrum, stereotypically perceived but also formally/diagnostically classified as having a mental disorder, are particularly vulnerable to testimonial injustice. Paul Crichton and colleagues point out a serious problem, which may in part cause testimonial injustice. The mere fact of having a mental illness or disorder seriously affects the credibility of a person’s testimony. Psychiatrists then tend to interpret the behaviors or explanations in question as the effects of the illness rather than as phenomena resulting from another cause. Crichton gives interesting examples from his own clinical practice:

“When one of the authors (P.C.) was a medical student in Munich, Germany, he saw a young man on an acute psychiatric ward who said he was a relative of the then Soviet leader. The responsible consultant took this to be a grandiose delusion, and therefore as evidence of a psychotic illness; it later turned out to be true”.[15] (p. 66)

People on the autism spectrum are often denied epistemic authority. Their testimony is denied credibility without justification, and they are not seen as fully-fledged epistemic agents [38]. This phenomenon occurs in two main ways in evaluations carried out by experts, such as medical examiners, diagnosticians, or forensic experts. It appears as the non-recognition or misinterpretation of the testimony of the service user and/or as the ignoring of the testimony of third parties involved (such as carers, who know the service user best or the opinions of the professional therapeutic teams working with the service user on a daily basis).

Suppose a person on the autism spectrum under assessment (or his/her carer) declares that he/she has severe difficulties in social relationships. This information can be easily trivialized by the expert, who has a stereotypical image of a person with autism as incapable of interpersonal relationships. Such an assessment, unfortunately, results in overlooking problems of a relational nature or due to other causes, e.g., depression, and neglecting therapy/treatment tailored to the person’s needs. Similar experiences of, inter alia, attributing social problems as resulting from diagnosed depression are described by Richard Lakeman in his studies on epistemic injustice in mental healthcare systems [11,39].

On the other hand, people on the autism spectrum are also subject to hermeneutical injustice. The media-disseminated stereotypical representations of autism that construct the social space of meanings and understandings of the spectrum can have positive and negative impacts. Grim’s observations can also be directly applied to the situation of people on the spectrum. Their accounts often mention that receiving a diagnosis allowed them to make sense of their behaviors or ways of thinking that they did not previously understand and for the interpretation of which they did not have good hermeneutic resources. The problem is that, if a negative or deficit message dominates socially, making sense of one’s experience leads to seeing oneself as ‘broken’ or one’s behavior as something shameful to be gotten rid of at all costs [40,41,42,43,44,45,46,47]. For example, most media coverage portrays people on the autism spectrum as suffering victims of their condition and shows autism as inferior to a neurotypical state and needing correction. People with autism internalize these views, which then become essential to their identities. Another example is the academic discourse, which primarily treats people with autism as objects of research rather than equal partners in the research programs.

As far as autism is concerned, the diagnosis and other kinds of expert assessment is problematic due to the lack of “hard data” (such as biological markers) and unequivocal findings as to its causes. The increasing public presence of autism, intensive scientific research, and growing media coverage often translate to the rise of numerous non-rational beliefs (such as stereotypes or prejudices) on this topic [33]. These beliefs might constitute the essential discursive presuppositions of cognitive errors concerning autism—including those made by the experts. As previous research showed, these presuppositions might even work at the level of our background knowledge [48]. The power relations between health experts and people with autism impact almost every area of autistic people’s lives, from the construction of their identities (based, e.g., on social, media, and professional representations), through interpersonal relationships at all levels of social organization, to decisions made by third parties related to their economic and social situation [49,50].

These discursive practices, which construe the social imagination and shape widely disseminated narrations, may lead to the reinforcing of hermeneutical epistemic injustice toward the users (and, as this paper argues further on, service providers) of the mental healthcare system:

‘(…) hermeneutical practices play an important role in health care because they allow service users sense-making reflectivity, which helps to turn a confusing and troubling set of symptoms into a more comprehensible and tenable context’.[3] (p. 169)

Thus, interpretive practices, based largely on socially available resources, are essential for autistic ‘patients’ meaning making concerning their lived experience. These practices are handicapped by hermeneutical injustice, e.g., when there are no other models for thinking of oneself than as a sick person or someone without a voice. Such tendencies are clearly visible in numerous media discourse analyses [41,44,46,51,52,53].

## 4. Mental Health Expertise and Its Fallibility

The concept of expertise is widely and thoroughly developed. There are overview monographs presenting the history and cultural context of knowledge formation of expertise [54], monographs analyzing various types of expertise [55], and monographs on selected types of expertise and their transformations [56,57]. Some studies show links of expertise to social [58] and political issues [59,60], as well as the recent changes in this phenomenon (e.g., declining trust in expertise, the crisis of expertise in a democratic society) [61].

One can distinguish two classical models of expert knowledge function, that is, ‘knowledge how’ and ‘knowledge that’ [62,63,64], and two aspects of expert knowledge, that is, the objective aspect (beliefs, assertions, rules for justifying them) and the subjective aspect (cognitive dispositions such as general cognitive abilities or epistemic virtues and expert competence). Both of these aspects are conditioned by several social factors. The concept of expertise and its applicable standards are shaped and transformed according to the social expectations and functions it is supposed to perform [65]. The subjective dispositions of experts are also not suspended in an information vacuum and are partly shaped by the discourses (professional, scientific, and popular) in which the expert in question functions.

From the perspective of social epistemology [66], expert knowledge is a set of primary and specialized information that comes from one or more scientific disciplines, remains at the expert’s disposal, and relates to the issue that is the subject of the formulated expert opinion. Notably, scientific knowledge is indirectly related to expert knowledge constituting its basis. However, the latter shows a higher degree of flexibility regarding the conditions for creating and changing the standards of reliability or credibility of the statements contained therein. These standards are much more susceptible to the influence of social, political, or everyday life factors [67,68].

The above is also true regarding specialized fields such as medicine and its subdisciplines, such as psychiatry. There is no ideal type of model of expert knowledge for all medical specializations. Instead, they function on account of a given practical situation, a specific problem situation, or a decision to be made. They express the demand of some group and respond to it, which all together determines a specific rather than a universal type of expertise.

Speaking of medicine, one should shift emphasis from expert knowledge towards expertise, understood as conditioned cognitive dispositions, such as thinking, deduction, justification, and argumentation. As Majdik and Keith put it:

‘“Expertise” as a concept cannot be reconciled by only one shared principle. As a consequence, it is not comprehensible in a conceptual definition but only in its varied uses and enactments. We suggest, in other words, that there is not—even deep down on a conceptual level—one kind of expertise, but kinds of expertise that resonate with kinds of problems’ [69]. This is clearly visible in psychiatry, where different disorders have particular diagnostic demands.

Experts in their fields might also be subject to stereotypes and simplified rules of reasoning [70,71,72,73,74]. Draaisma [75] and Hacking [76,77] showed that even domain-specific experts are not fully stereotype-proof and might be influenced by simplified inferencing patterns (the so-called heuristics).

Mental health experts, such as psychiatrists and clinical psychologists, are especially exposed to systemic cognitive errors [78] as mental states, disorders, or illnesses are difficult to measure in an objectified way. Experts must contend with the co-existence of various conditions, including somatic ones, with considerable personal differences, lack of biomarkers, or non-specific symptoms. They also often base their judgments on intuitions whose reliability is unclear [79,80].

Naturalistic decision-making theory states that heuristics allow experts to make complex decisions quickly [70,72,81,82] when expert opinions, e.g., in a case-law process, must be issued mainly based on intuitive decisions [71,81,82]. On the other hand, the heuristics and biases approach theory sees heuristics as a source of cognitive errors [72]. Both approaches emphasise that heuristics are inherent parts of our decision-making processes. We cannot “turn them off”. We may only try to become aware of them and control their effects.

Damasio [83] and Ericsson [84] point out that despite the differences, both theories indicate three primary conditions required for correct intuitive decisions: (1) update knowledge based on credible sources in a specific field; (2) deliberate practice (systematically applying expertise continuously subjected to reflection); (3) feedback on the decisions made. If these conditions are not met, experts might be prone to mistakes and their elimination [48].

## 5. Epistemic Virtues of Experts

Two modes of epistemic injustice can compromise the professional traits and qualities that experts are typically expected to possess. In performing their duties, experts form numerous opinions, make judgements or diagnoses based on their beliefs. Ideally, these beliefs should be (a) always true, (b) formulated based on a complete set of information obtained from all available sources, (c) supported by relevant evidence [85], (d) free from systemic biases, and (e) subjected to multi-stage control.

Unfortunately, in the case of mental health phenomena, there is often (a) no single, universally valid description of a given reality that we can define as ‘true’, (b) sources of information are sometimes incomplete (some of them are also ignored by experts for various reasons) and the information itself is usually subject to a process of interpretation, (c) there is usually no ‘hard’ evidence, (d) experts are not free from biases, (e) often their decisions are not subject to automatic review—which is a result of the trust we place in those who have, sometimes rightly sometimes wrongly, the appropriate epistemic authority [86,87,88].

In such a situation, to avoid more or less chaotic and ‘luck-based’ beliefs and decisions [89,90,91], expert knowledge theory should focus not solely on ‘the outcome’ but on the process of belief justification. According to the externalist view of Alvin Goldman’s epistemological reliabilism [92,93,94], the most crucial feature of the knowledge creation process is the reliability of cognitive and epistemic processes. It means that ‘what makes a belief epistemically justified is the cognitive reliability of the causal process via which it was produced’ [89].

Building on the findings of the reliabilists, virtue epistemologists (Ernst Sosa [90,91,95,96], John Greco [97] John Turri [98]) take the view that whether the process of obtaining a belief has been carried out reliably is determined to the greatest extent by the characteristics of the subject himself (the notion of a reliable process is replaced by concepts such as ‘competence’, ‘virtue’, ‘skill’ or ‘ability’ [99,100]).

Virtue epistemologists, however, strongly emphasize the subject’s role in justifying beliefs and creating knowledge. The epistemic virtues of the subject, such as reliability, open-mindedness, understanding of another point of view, or intellectual humility, must form the basis for justifying the subject’s beliefs. Reliable expert knowledge results from a cognitive process in which the subject can use his or her epistemic virtues in a controlled manner.

For example, a list of such epistemic virtues includes intellectual conscientiousness, open-mindedness, attentiveness, curiosity, discernment, humility, objectivity, and understanding. The list of epistemic vices includes dogmatism, epistemic blindness, intellectual dishonesty, self-deception, the superficiality of thought, and superstition. Many such vices are responsible for the aforementioned cognitive biases [101].

The knowledge basis for experts’ decisions is ‘true belief out of intellectual virtue, a belief that turns out right because of the virtue and not just by coincidence’ [90] (p. 277). Therefore, any process that might negatively influence such experts’ virtues and hermeneutical abilities might be considered harmful. The phenomenon of epistemic injustice is such a case. Virtue theorists think cognitive biases result from over-reliance on intuitive, fast, non-discursive, and non-reflective forms of cognition. In their view, this deficiency might be controlled ‘by cultivating responsibilist virtues, such as epistemic humility and self-vigilance’ [102] (p. 2).

## 6. Epistemic Injustice toward Mental Health Experts

The up-to-date discussions have focused on epistemic injustice concerning service users. The sources of such epistemic injustice lie within the system of mental health experts and their judgments, and we have described its two subtypes regarding people on the autism spectrum. In the power dynamics of autistic and non-autistic people, the focus remained on how epistemically unjust expert decisions influence the patients’ quality of life and self-esteem, among others. In short, it is related to the expert-to-patient transfer of “injustice”.

Taking into account the tasks set for experts (giving opinions and decisions that are factual, reliable, and conforming to the best standards) and their dispositions, abilities, and epistemic virtues, it is worth looking at experts, usually shown as perpetrators of epistemic injustice as its victims (see Table 1).

In what sense and why are experts the object of epistemic injustice? Two additional questions are due. First, what is the harm resulting from cognitive injustice? Second, who perpetrates this injustice against experts?

Following the reliabilist theories of expert knowledge, the harm consists primarily in the deformation and negative impact on experts’ dispositions and competencies. Consequently, it hinders and even prevents the efficient and reliable performance of professional tasks.

The harm is different in the case of testimonial injustice. The testimony of one expert may be undermined or rejected by another. It thus loses its justifying force regarding an opinion or diagnosis. Such situations occur when issuing interdisciplinary diagnoses or at the committee assessment of the degree of disability of a patient. An opinion-issuing expert should be guided by the patient’s medical documentation (other ‘experts’ opinions). It may happen that, in whole or in part, such an opinion is not considered. This may also result from the internally assumed hierarchy of epistemic credibility within the health care system. For example, an opinion of a psychiatrist is usually perceived as more credible than a psychotherapist’s.

Another example of testimonial injustice harm is disregarding experts’ intuitive decision-making, such as a rapid diagnosis of schizophrenia based on the so-called Praecox Feeling [79,80,102,103]. It has been known and theorized since Ruemke that qualified psychiatrists can pronounce adequate judgments of schizophrenia based on intuition alone—a theory that has some experimental, although not conclusive, confirmation [80,103,104,105,106]. These judgments are most likely based on immediate and pre-reflective intercorporal affective exchange with the patients, which does not present itself as a set of operationalized symptoms described by diagnostic manuals. Since current psychiatric diagnoses must be reliable, psychiatrists cannot openly speak of their intuitions, which would likely undermine their professional credibility. Instead, this sometimes vital aspect of the diagnostic process must remain publicly hidden. The psychiatrists’ testimony regarding the source of their knowledge is, therefore, unjustly silenced by the dominant discourse of diagnostic tools operating within the “ticking boxes” framework [107]. In both cases, others do not recognize or value the expertise. It can be dismissed by those in power, for example, psychiatrists over counselors, or may be blocked by the functional aspects of the medical discourse, such as formalized disregard of clinical intuition.

The nature of the harm is different when experts are subject to hermeneutical injustice. Again, we speak of hermeneutical injustice “when there is a breach in shared conceptual, interpretative resources that puts individuals at a disadvantage when trying to make sense of their experiences” [3] (p. 158). To use the above example once more, a psychiatrist himself may disregard one’s diagnostic intuition due to being embedded in the dominant discourse that does not recognize its value.

In the case of mental health experts, the hermeneutical injustice category includes biased or limited preconceptions about expert’s own position and its capabilities and also extends to interpreting the experience of service users assessed. For example, an expert psychiatrist rooted firmly in the dominant biomedical tradition may find it difficult to perceive and understand the psycho-social difficulties of patients due to perceiving and interpreting oneself as normatively “normal”. Moreover, not being entirely immune to the numerous cognitive errors based on popular stereotypes and simplifications present in various discourses, his/her ability to make a competent judgment about a patient’s condition, and level of functioning may be compromised. In this sense, epistemic injustice is also dialectical and may operate between both parties.

Hermeneutical injustice thus concerns medical experts when they are unable to fully understand or participate in the autism discourse due to a lack of shared interpretive resources, e.g., when they are unable to effectively communicate with or understand patients’ perspectives due to a lack of shared language or cultural understanding. This can lead to misdiagnosis or inadequate treatment, as they may not fully understand the patient’s symptoms or concerns. It can also lead to a lack of trust and credibility, as the patients may feel that their concerns are not being fully heard or understood.

## 7. The Sources of Epistemic Injustice toward Experts

There are several answers to the question of who perpetrates epistemic injustice against experts.

Firstly, testimonial injustice occurs when one expert refuses to consider another expert’s opinion as relevant. This problem can be encountered, for example, during court hearings when expert witnesses are called upon to assess a case. It may also occur during committees deciding on the degree of disability and state assistance. An expert usually receives therapeutic documentation that includes the opinions of other experts working with the patient in question. These opinions may be ignored as not containing substantively relevant arguments. Perhaps this is due to acceptable ways of valuing evidence. As Crichton and colleagues put it:

“Health professionals are trained to place higher value on ‘hard’ or objective evidence, namely the results of investigations, than on ‘soft’ or subjective evidence provided by patients. In psychiatry, there is virtually no hard evidence and diagnoses have to be made mainly on the basis of what patients say and how they behave”.[15] (p. 67)

A similar problem occurs when ignoring not only the opinion of patients but also of other experts. In the case of mental conditions, one is usually not dealing with ‘hard’ evidence but with the opinions of others (even if they are other experts). The decision-making expert may put the observed condition of the patient above the written opinions received. However, it may be the case that the difficulties faced by people on the autism spectrum (such as difficulties of a relational and social nature) cannot be observed during a brief assessment, especially when dealing with individuals who camouflage their difficulties very well. This may lead to a cognitive error called WYSIATI—what you see is all that is. WYSIATI thus becomes a source of epistemic injustice towards another expert [71].

As a result, some experts may suffer unequal opportunities to contribute to their field and not be equally represented in the decision-making bodies. In consequence, the voices of some experts may not be heard analogically to the voices of patients with autism. Furthermore, such a situation might exacerbate the lack of diversity within the medical profession and marginalization of community experts facing discrimination in their professional judgments.

Secondly, more hermeneutic types of epistemic injustice may stem from traditions and cultural and systemic conditions in which experts gain knowledge, experience, and operate.

The biomedical model, increasingly criticized but still dominant in psychiatry today, is one of the most important traditions shaping today’s understanding of such phenomena as autism. Health professionals are attached to this model through education, subsequent practice, and the lack of viable alternatives on a systemic scale. It is through its prism that phenomena having a largely psycho-social background are assessed. In practice, this means that experts judge people on the autism spectrum through their somatic or physical difficulties (performing basic self-care activities or sensory hypersensitivity) instead of difficulties in social relations or the emotional sphere, which often are fundamental.

Furthermore, as David Peña-Guzmán and Joel Reynolds showed:

‘(…) medical experts are also part of an institution with a long and dark history concerning disability. Historically, medicine has played a central role in the construction of disability as both spectacle and tragedy (…)’.[108] (p. 223)

This grounds communicative difficulties between disabled or mentally ill patients’ communities and medical professionals. This state of affairs persists and is still reproduced in many opinion-forming discourses at the media and government levels [24,44,45,109]. An analysis of psychoanalytic discourse on autistic patients showed that, although professionals try to maintain a commendable intellectual humility and break with many popular narratives concerning the nature and origins of autism, they still return to the dominant rhetoric of deficit and unhappiness that autism advocacy communities fight today [110].

Medical professionals’ reliance on presupposed epistemic schemes (such as ableism belief in the rare occurrence of autism in females or overdiagnosis of autism due to media interest) may lead to misdiagnosis and disrupt the flow of information needed for effective diagnosis or treatment planning.

These are examples of ‘breach in shared conceptual and interpretative resources’ that can be the basis for hermeneutical injustice. Rooted in certain intellectual traditions and the lack of alternatives, these schemes may significantly affect ‘experts’ self-understanding and hinder their reliable performance of their duties.

Hermeneutical injustice concerning the diagnosis of autism thus has two dialectical sides, so to speak, and one does not exist without the other. It is crucial for autism experts to be mindful of this potential for hermeneutical injustice and to work to actively address it by engaging in inclusive and collaborative practices that seek to understand and respect the perspectives of autistic individuals. This can include consulting with and listening to people on the spectrum, seeking out diverse perspectives and experiences, and actively working to dismantle any barriers that may prevent autistic people from fully participating in and contributing to discussions and decisions that affect them.

Peña-Guzmán and Reynolds seem to capture the root of this problem with regard to the relationship between medical professionals and disability communities, and patients:

‘We submit that at the root of these mechanisms is the medical community’s lack of engagement with critical, non-medical modes of knowledge concerning disability, including and especially with respect to knowledge created by disability communities themselves, as well as bodies of work which draw directly on such knowledge, as literature in disability studies and philosophy of disability regularly does. In other words, a root cause of ableism in medicine is medicine’s understanding of disability as an objective lack rather than as a diverse set of phenomena that are thoroughly socially mediated’.[108] (p. 225)

The third source of epistemic injustice is popular discourses, media, academic, or scientific, of which experts by experience are a part. As far as autism is concerned, the media disseminate the image of people deprived of the ability to reason and, to a significant extent, self-determination. It is the image of people suffering because of their autism and not from the social environment not understanding their specific needs.

More and more, but still insufficiently, attention is paid to experts by experience in various discourses. The overlapping scientific and media discourses radically marginalize the first-person perspective and the voice of people on the autism spectrum. Even when this voice does appear, it is usually not accorded proper epistemic authority. Decision-making theories indicate that when experts make quick and complex, intuitive decisions, they are exposed to the influence of stereotypical depictions of various mental conditions from media or outdated scientific discourse. One of the epistemic schemes used is the systemic marginalization of the voice of the mentally ill. Our previous empirical study showed how deeply rooted in background knowledge the preconceptions concerning autism might be, despite the high declarative quality of knowledge [111,112]. According to Heggen and Berg:

‘Epistemic injustice can be a consequence of low disease prestige and negative stereotypes leading to bias against the knower or privileging certain epistemic and practice ideals like EBP, or privileging knowledge derived from medical training and theory. Health personnel might have the very best intentions to trust a patient and believe what the patient is telling them but nevertheless ignore the patient’s testimony, for instance because it is not in accordance with medical expertise. Consequently, patients testimonies are not considered credible, and they are undermined as first-hand knowers. Their reports about their condition are marginalized during medical examination and they encounter difficulties in their efforts to make themselves understood’.[12] (p. 3)

It is difficult to consider the social role played by experts and the social conditions for producing and using expert knowledge in isolation from ethical issues. The epistemic authority that allows a person to be referred to as an expert is irrevocably connected with adequate fulfillment of such a role. Following virtue epistemology, an expert should justify his/her beliefs reliably to make it possible. S/he should use expert-appropriate dispositions, and precisely those, which are often disturbed resulting in epistemic injustice.

## Figures and Tables

**Table 1 diagnostics-13-00927-t001:** Epistemic injustice as affecting both service users and mental health experts.

	Testimonial Injustice	Hermeneutical Injustice
Sources of epistemic injustice	- low level of intellectual humility - epistemic ignorance - outdated but still dominating discourses - hierarchy of epistemic authorities within mental healthcare systems	- outdated but still dominating discourses - lack of critical evaluation of possessed knowledge and paradigms utilized
Service users as objects of epistemic injustice	- ignored as knowers and denied the epistemic authority - insufficient degree of input in one’s diagnostic or therapeutical process	- limited ability to interpret one’s own experience outside of dominative, often harming, narratives
Mental health experts as objects of epistemic injustice	- ignored as a reliable source of knowledge by other experts and thus excluded from knowledge creation processes - unable to utilize a full potential of information sources in the decision-making processes	- limited ability to understand patient’s point of view - biased preconceptions about patient’s condition - biased or limited preconceptions about their own expert position and its capabilities - sanctioned by traditional background assumptions concerning the nature of mental health phenomena

## Data Availability

Not applicable.

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
