# Peer review of "Mental Health Experts as Objects of Epistemic Injustice—The Case of Autism Spectrum Condition"

_diagnostics, 2023, doi:10.3390/diagnostics13050927_

Round 1
Reviewer 1 Report
This paper is an interesting contribution that complements the biomedical studies on diagnosis. Its perspective is philosophical and, humbly, I think that gives it an added value that should be taken into account. In fact, there are also sociological works of great interest for studies on biomedical diagnosis.
The work is very clear and well structured. It is well understood and the documentation is exhaustive. It is especially interesting that the authors introduce hermeneutics and Ernest Sosa's proposal, I agree on the relationship between the two. All this adds great value to the study. In fact, the article is profound and shows a great knowledge of the subject. However, this also implies some problems for a non-specialized reader.
Well... in my opinion, some minor adjustments should be made to help the reader who does not know much about the subject. The improvements that I think should be made are the following:
1) It would be convenient clarify the concept of epistemic injustice in the introduction, since it is not clear who exercises such injustice. In other words, it would be good to clarify in a concrete way who could exercise epistemic injustice in the health system. In my opinion, any person: physicians, nurses and even family members could be involved.
2) Similar to the above, I think the concept of dominant narratives should be clarified a bit. I'm not sure if everyone would understand it.
3) Table 1 would look better if it were in its entirety on page 8.
4) There are errors in the format of the article, they should be revised.
5) There are errors in the bibliographic references, for example 70 is wrong and in others there is no page or article number.
Further comments:
With a view to future research in this line of work, I would like to suggest to the authors that they look for sociological texts. There are currently a multitude of works on the social determinants of diagnosis and how these affect the physician and the patient. I am aware that the authors have carried out a philosophical study, but I humbly believe that it would help them to better understand and study other unconsidered aspects of epistemic injustice.
Author Response
Response to Reviewer 1 Comments
Thank you for your helpful and useful comments. We have taken all of them into account. Below we respond point by point in how we have addressed them.
Point 1: It would be convenient clarify the concept of epistemic injustice in the Introduction, since it is not clear who exercises such injustice. In other words, it would be good to clarify in a concrete way who could exercise epistemic injustice in the health system. In my opinion, any person: physicians, nurses and even family members could be involved.
Similar to the above, I think the concept of dominant narratives should be clarified a bit. I'm not sure if everyone would understand it.
Response 1: Thank you very much for this comment. As suggested, we have added a paragraph in the Introduction clarifying who can perpetrate cognitive injustice in the meaning we are discussing. We have also explained in the next section ('Epistemic injustice towards people on the autism spectrum') what we mean by the term main narratives and gave examples.
Point 2: Table 1 would look better if it were in its entirety on page 8.
Response 2: As suggested by Reviewer #4, we have moved the table to a different place in the article, but its final position will depend on the arrangement of all elements in the finalized manuscript file prepared by the editorial office.
Point 3: There are errors in the format of the article, they should be revised.
Response 3: We have carefully checked the text and made corrections in this respect.
Point 4: There are errors in the bibliographic references, for example 70 is wrong and in others there is no page or article number.
Response 4: After adding a few new items to the reference list resulting from amendments made to the text, we reviewed the bibliography carefully and removed all shortcomings.
Point 5: Further comments:
With a view to future research in this line of work, I would like to suggest to the authors that they look for sociological texts. There are currently a multitude of works on the social determinants of diagnosis and how these affect the physician and the patient. I am aware that the authors have carried out a philosophical study, but I humbly believe that it would help them to better understand and study other unconsidered aspects of epistemic injustice.
Response 5: Thank you very much for this suggestion. Indeed, in the current article, we have focused primarily on the philosophical perspective, but in the future, as we continue to work on this issue, we will certainly make more use of the literature and the sociological perspective on the subject.
Reviewer 2 Report
Epistemic injustice, a concept developed by philosopher Miranda Fricker, is a situation in which, without adequate reason, someone suffers a harm caused by or related to access to, the process of producing and distributing knowledge. This paper presents the issue of epistemic injustice in a novel perspective. Although the literature describes many possible types of epistemic injustice, this paper focuses on its two main modes as described by Fricker, namely testimonial and hermeneutical injustice. Obtaining the epistemic authority that allows a person to be referred to as an expert is irrevocably connected with the issue of adequate fulfillment of such a role. For this to be possible, an expert - following the views of virtue epistemologists cited earlier - should justify their beliefs reliably, using dispositions appropriate for 461 experts, which are often disturbed by numerous environmental, social, and subjective factors leading to experts epistemic injustice. I find this contribution of interest to a wide audience.
Author Response
Response to Reviewer 2 Comments
Review: Epistemic injustice, a concept developed by philosopher Miranda Fricker, is a situation in which, without adequate reason, someone suffers a harm caused by or related to access to, the process of producing and distributing knowledge. This paper presents the issue of epistemic injustice in a novel perspective. Although the literature describes many possible types of epistemic injustice, this paper focuses on its two main modes as described by Fricker, namely testimonial and hermeneutical injustice. Obtaining the epistemic authority that allows a person to be referred to as an expert is irrevocably connected with the issue of adequate fulfillment of such a role. For this to be possible, an expert - following the views of virtue epistemologists cited earlier - should justify their beliefs reliably, using dispositions appropriate for 461 experts, which are often disturbed by numerous environmental, social, and subjective factors leading to experts epistemic injustice. I find this contribution of interest to a wide audience.
Response 1: Thank you for taking the time and effort to review our manuscript. We are pleased that it was considered as a meaningful contribution to the field.
Reviewer 3 Report
This is an interesting article, but major revision is required in order to improve it:
The aim of the paper must be clearly presented. There is no section with the relevant research questions/hypotheses.
A more detailed search of theoretical and research papers must be made. Relevant research has to be mentioned such the following among others that examines a widespread (but recently examined relevant with epistemic injustice concept/phenomenon) that of hubris in healthcare as a form of injustice:
Owen, D. (2008). Hubris syndrome. Clinical Medicine, 8(4), 428.
Owen, L. D. (2006). Hubris and nemesis in heads of government. Journal of the Royal Society of Medicine, 99(11), 548-551.
Giannouli, V. (2017). What do we really know about hubris, culture and health professionals in leadership positions? A methodological recommendation. Asian journal of psychiatry, 26, 150-151.
Giannouli, V. (2021). Exploring hubris in physicians: Are there emotional correlates?. Psychiatria Danubina, 33(1), 57-59.
Lincoln, M. (2020). Study the role of hubris in nations' COVID-19 response. Nature, 585(7825), 325-326.
Vogelstein, E. (2016). Professional Hubris and its Consequences: Why Organizations of Health‐Care Professions Should Not Adopt Ethically Controversial Positions. Bioethics, 30(4), 234-243.
Another useful point to mention regading epistemic injustice can be found in a discussion:
In addition to the above, there should be a reframing of the table, so it is more easy to grasp.
Author Response
Response to Reviewer 3 Comments
Thank you for your helpful and useful comments. We have taken all of them into account. Below we respond point by point in how we have addressed them.
Point 1: The aim of the paper must be clearly presented. There is no section with the relevant research questions/hypotheses.
Response 1: In the Introduction section, we have added an opening paragraph outlining the research questions we are posing and answering in the article, and sketched the main hypothesis. Due to the type of article (Review), we have refrained from introducing a separate section on research questions and hypotheses.
Point 2: A more detailed search of theoretical and research papers must be made. Relevant research has to be mentioned such the following among others that examines a widespread (but recently examined relevant with epistemic injustice concept/phenomenon) that of hubris in healthcare as a form of injustice:
Owen, D. (2008). Hubris syndrome. Clinical Medicine, 8(4), 428.
Owen, L. D. (2006). Hubris and nemesis in heads of government. Journal of the Royal Society of Medicine, 99(11), 548-551.
Giannouli, V. (2017). What do we really know about hubris, culture and health professionals in leadership positions? A methodological recommendation. Asian journal of psychiatry, 26, 150-151.
Giannouli, V. (2021). Exploring hubris in physicians: Are there emotional correlates?. Psychiatria Danubina, 33(1), 57-59.
Lincoln, M. (2020). Study the role of hubris in nations' COVID-19 response. Nature, 585(7825), 325-326.
Vogelstein, E. (2016). Professional Hubris and its Consequences: Why Organizations of Health‐Care Professions Should Not Adopt Ethically Controversial Positions. Bioethics, 30(4), 234-243.
Response 2: Thank you for this important suggestion. We have added 3 paragraphs in the second section of the article (‘Epistemic injustice towards patients’) mentioning and shortly discussing a spectrum of contemporary research on the manifestations of epistemic injustice, its causes and potential remedies (including some of the studies suggested by the Reviewer):
‘Epistemic inequalities result in ethnic minorities being forced to use crisis care and are less likely to receive primary mental health care (Bansal et.al. 2022). The differences between patients' first-person perspectives on illness and service providers' conceptualization of illness as a disease are growing (Heggen, Berg 2021). Patients' testimonies of perceived illness symptoms are downgraded (Tosas 2021) and somatic illnesses are often overshadowed by previously detected mental conditions leading to a decrease in the quality of medical care (Bueter).
Also, epistemic injustice impacts the policy-making level resulting in the exclusion of specific patients or health professionals groups from the legislative processes (Michaels 2021, Drozdzowicz 2021).
Exploring the ideas such as epistemic solidarity among patients or increasing the role of biocommunities in the process of knowledge production and circulation might reduce epistemic injustice in mental health (Pot 2022, Wodziński, Rządeczka, Moskalewicz 2022). Also Eequally important is the identification of causes and additional factors influencing the occurrence of epistemic injustice, such as the hubris phenomenon (Owen 2008, Vogelstein 2016) or low levels of intellectual humility among professionals (Bak, Kutnik 2021), which increase the risk of systemic cognitive errors.’
Point 3: In addition to the above, there should be a reframing of the table, so it is more easy to grasp.
Response 3: We have restructured the table adapting it to the suggestion made by the Reviewer #4.
Reviewer 4 Report
The manuscript "Mental Health Experts as Objects of Epistemic Injustice" is an important narrative review on the multifaceted challenges of psychiatry and clinical psychology. In contrast to somatic disorders mental disorders rarely have biomarkers and are often overlapping (see RDoC) and subjectivity reigns.
Historically mental disorders have been perceived differently than what we know now about their "causes".
The authors review the epistemic injustice although the paper would greatly benefit from some restructuring and clearer structure.
Table 1 should come earlier, and the last row be the first. Then - conditioned on what the sources of injustice are - the article should be structured by how this affects
a) the patients - case of autism spectrum disorder (perceiving as faulty instead of neurodiverse)
b) the clinician
c) the interaction between patient and clinician (cure vs accepting and coping)
d) the scientific field / society (view of it as disorder vs neurodiversity)
this would help the reader and provide the arguments in a more systematic way.
The title should include "the case of autism spectrum disorder" or smth similar
minor:
line 190 "perspectives agree that heuristics is indispensable" "is" should be "are"
some spacing issues
Author Response
Response to Reviewer 4 Comments
Thank you for your helpful and useful comments. We have taken all of them into account. Below we respond point by point in how we have addressed them.
The authors review the epistemic injustice although the paper would greatly benefit from some restructuring and clearer structure.
Point 1: Table 1 should come earlier, and the last row be the first.
Response 1: Following the reviewer's suggestion, we have moved the table to the first section of the article, so that its content anticipates the information appearing in the text. We have also changed the order of the rows as suggested.
Point 2: Then - conditioned on what the sources of injustice are - the article should be structured by how this affects
- a) the patients - case of autism spectrum disorder (perceiving as faulty instead of neurodiverse)
- b) the clinician
- c) the interaction between patient and clinician (cure vs accepting and coping)
- d) the scientific field / society (view of it as disorder vs neurodiversity)
this would help the reader and provide the arguments in a more systematic way.
Response 2: Thank you in particular for this comment, as it allowed us to make the argument of our article clearer. We have changed most of the headings and moved some larger sections of the text, so that they form a structure similar to the one proposed by the Reviewer (these changes are visible in the track changes mode). The article now 1) introduces and discusses in more detail the concept of epistemic injustice in the Introduction; 2) describes the phenomenon of epistemic injustice towards patients; 3) outlines the context of mental health expertise and discusses the issue of epistemic virtues; 4) describes epistemic injustice towards experts in more detail; 5) identifies and discusses the sources of epistemic injustice towards experts.
With the suggested changes, the issues addressed are kept separate and do not get mixed up with each other.
Point 3: The title should include "the case of autism spectrum disorder" or smth similar
Response 3: We have added the sub-heading suggested by the Reviewer.
Point 4: line 190 "perspectives agree that heuristics is indispensable" "is" should be "are"
some spacing issues
Response 4: We have made the indicated changes and several other improvements related to the language used and the formatting of the article (visible in track changes mode).